# CASE: Commonsense-Augmented Score with an Expanded Answer Space

**Wenkai Chen** and **Sahithya Ravi** and **Vered Shwartz**
University of British Columbia
Vector Institute for AI
{wkchen, sahiravi, vshwartz}@cs.ubc.ca

## Abstract

LLMs have demonstrated impressive zero-shot performance on NLP tasks thanks to the knowledge they acquired in their training. In multiple-choice QA tasks, the LM probabilities are used as an imperfect measure of the plausibility of each answer choice. One of the major limitations of the basic score is that it treats all words as equally important. We propose CASE, a Commonsense-Augmented Score with an Expanded Answer Space. CASE addresses this limitation by assigning importance weights for individual words based on their semantic relations to other words in the input. The dynamic weighting approach outperforms basic LM scores, not only because it reduces noise from unimportant words, but also because it informs the model of implicit commonsense knowledge that may be useful for answering the question. We then also follow prior work in expanding the answer space by generating lexically-divergent answers that are conceptually-similar to the choices. When combined with answer space expansion, our method outperforms strong baselines on 5 commonsense benchmarks. We further show these two approaches are complementary and may be especially beneficial when using smaller LMs.

## 1 Introduction

Large language models (LLMs) have demonstrated strong few-shot and zero-shot performance across various NLP tasks, with the larger models often matching earlier fine-tuned approaches that relied on task-specific labeled data (Radford et al., 2019; Brown et al., 2020a; Touvron et al., 2023). We focus on the zero-shot setup, which assumes that the knowledge needed to perform a specific task is already present in the LLM (Petroni et al., 2019; Zhou et al., 2020; Saha et al., 2022). Zero-shot learning has been employed for tasks such as translating between unseen language pairs (Zhang et al., 2020), summarization (Brown et al., 2020a), commonsense reasoning (Shwartz et al., 2020; Klein

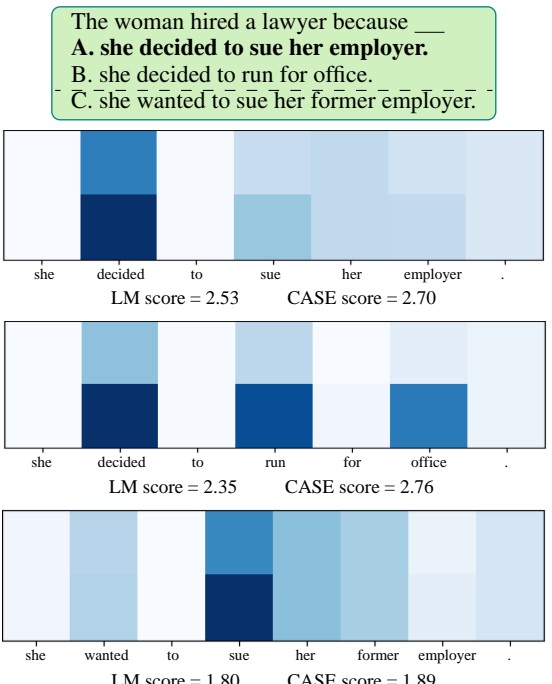

Figure 1: An example from COPA. A and B are the original options, while option C was generated by GPT-2 as part of the answer space expansion step. The top line in each heatmap represent the LM (cross-entropy) score and the bottom line represents our CASE score. Higher scores and blue blocks correspond to lower plausibility. CASE correctly predicts option A (and option C which is an expansion of A) as more plausible than option B, while the LM-score incorrectly predicts option B.

and Nabi, 2021; Liu et al., 2022; Fang et al., 2022), and more.

In multiple-choice question answering (MCQA) tasks, zero-shot methods typically rely on the language model (LM) probabilities as a proxy for plausibility, predicting the answer choice with the highest probability conditioned on the question. LM score is a naïve proxy for plausibility, since it confounds factors such as length, unigram frequency, and more (Holtzman et al., 2021; Niu et al., 2021). Indeed, in Figure 1, a GPT-2 based LM score incorrectly predicts that the woman hired a lawyer

because she decided to run for office, rather than because she decided to sue her employer.

In this paper, we propose to address one of the major limitations of the LM score. By summing or averaging the token-level probabilities, the LM score treats all tokens as equally important. A person reading this question would likely pay attention to option A because the word "sue" is highly relevant in the context of a lawyer. This signal might be weaker in a basic LM score where the word "sue" is conditioned on each other token in the question and previous tokens in the answer. Furthermore, the LM might miss non-trivial connections between related words.

To address this challenge, we propose CASE: a **C**ommonsense-**A**ugmented **S**core with an **E**xpanded Answer Space. CASE is a post-hoc dynamic weight scoring algorithm that prioritizes important words in the sentence. The importance of each individual word is determined based on its relationship with other words in ConceptNet (Speer et al., 2017). For example, ConceptNet provides the information that "sue requires having a lawyer". We use the word-level importance scores to re-weigh the LM probability scores. Indeed, in the second line of option A in Figure 1, the importance of the word "sue" increases the score of the entire sentence, leading to correctly predicting A as the correct answer.

We further adopt the strategy suggested by Niu et al. (2021) to expand the answer space by using a LM to generate additional answers and then mapping semantically-similar generated answers into the original space. This mitigates the LM score's sensitivity to infrequent words. Figure 1 demonstrates that a generated option C, "she wanted to sue her former employer", which is conceptually similar to A, further yields a higher probability score with our method.

We tested CASE on 5 popular commonsense MCQA datasets. CASE outperformed the broad range of strong baselines that we compared with, confirming that it is an effective method for zero-shot MCQA. We further study the impact of different model sizes, answer candidates of varying qualities, and different weight assignment strategies on the performance.[1]

---

[1]Our code is available at Github.

## 2 Background

### 2.1 Plausibility Scoring

Although the plausibility score of a sentence can be easily calculated by accumulating the probability assigned by the LM for each token, this approach suffers from various statistical biases such as sensitivity to the number of tokens, subword tokenization, and word frequency (Abdou et al., 2020; Holtzman et al., 2021). To address these biases, several improvements have been proposed. With respect to the length bias, prior work normalized the score by length (Mao et al., 2019; Brown et al., 2020b), or focused on the conditional probabilities of the question, which unlike the answer choices has a fixed length (Trinh and Le, 2018; Tamborrino et al., 2020). To factor out word frequency, Holtzman et al. (2021) proposed Domain Conditional Pointwise Mutual Information (DCPMI), which normalizes the conditional probability of the answer given the question by the prior probability of the answer. This is computed as the conditional probability of the answer given a domain-specific prefix such as "The sentiment of the movie is" for sentiment analysis or "The answer is" for general QA tasks. SEQA (Niu et al., 2021) mitigates the sensitivity to word choice by generating answers using GPT-2, and selecting the answer choice most similar to the generated answers.

Existing methods solely focus on the relationship between words in the choices and words in the question, ignoring the importance of each word for the decision. In this paper, we propose a new token-level weighting method to consider the importance of different words within the sentence based on their relationship to other words.

### 2.2 Knowledge-Enhanced Models

Zero-shot LM-based scoring methods implicitly reason about which answer is more likely based on the token-level probabilities. However, many tasks require multiple steps of reasoning to reach the correct answer (e.g., Mihaylov et al., 2018; Yang et al., 2018; Khot et al., 2020). A common approach is to retrieve relevant commonsense knowledge from knowledge bases (KBs) such as ConceptNet (Speer et al., 2017) and ATOMIC (Sap et al., 2019a; Hwang et al., 2021), in order to enhance the neural model and explicate the reasoning steps (e.g., Bauer et al., 2018; Xia et al., 2019; Lin et al., 2019; Guan et al., 2019; Chen et al., 2020; Huang et al., 2021). More recent work used the COMET model

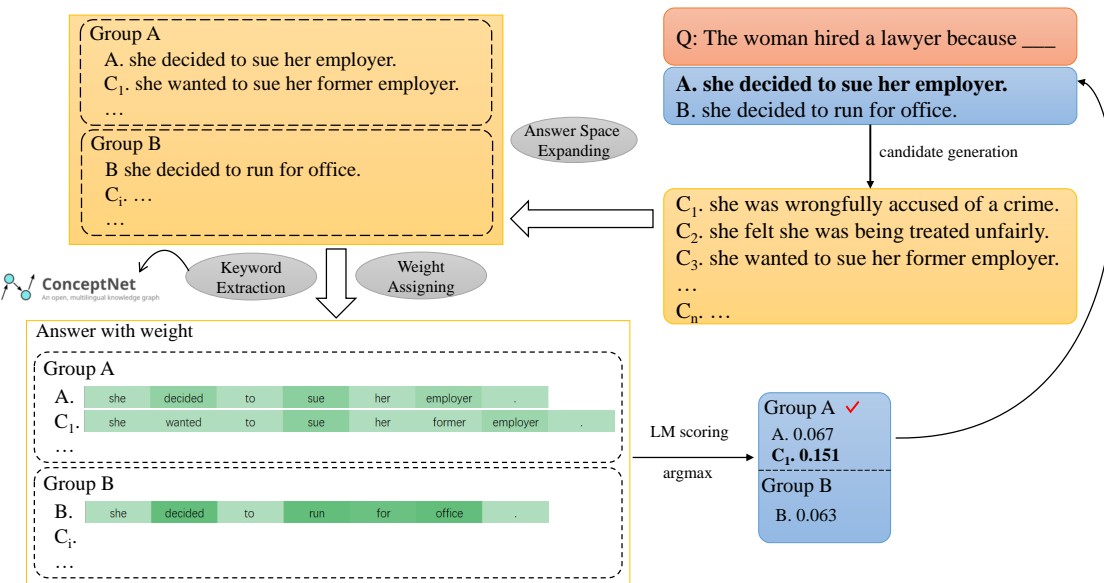

Figure 2: Overview of CASE, illustrated with an example from the COPA dataset. Groups A and B correspond to original choices A and B and any generated answers mapped to them (§3.3). Each word in each answer is scored based on its ConceptNet relationships to other words in the instance (§3.2). The score for each answer is based on the word probabilities (§3.1), weighted by the word-level scores. Finally, CASE predicts the answer choice with the highest scoring answer in its group.

(Bosselut et al., 2019; Hwang et al., 2021), which is a LM fine-tuned on the aforementioned KBs, to enhance models with high-coverage contextualized commonsense inferences (e.g., Majumder et al., 2020; Bosselut et al., 2021; Kim et al., 2022; Chakrabarty et al., 2022; Ravi et al., 2023).

An alternative recent approach which doesn't rely on external KBs prompts a LM to generate additional knowledge which is then incorporated back into the LM to make the prediction. Shwartz et al. (2020) and later Liu et al. (2022) used a LM to generate questions and answers about an MCQA instance. The answers to the questions are then incorporated into the LM-based scoring model as additional knowledge. Wei et al. (2022) proposed the popular chain-of-thought (COT) prompting approach in which the LM is taught through examples to generate multiple steps of reasoning followed by the answer to the question. In the zero-shot version, the LM is instructed to "think step-by-step". Finally, following concerns about the faithfulness of CoT inferences, Creswell et al. (2022) proposed to iteratively select parts of the inputs and draw inferences on them.

## 3 Method

We propose CASE, a **C**ommonsense-**A**ugmented **S**coring method with an **E**xpanded Answer Space.

CASE can be used for zero-shot MCQA tasks. It is based on LM score (Section 3.1). However, rather than treating all words in the context and answers as equally important, we propose a weighted score where the conditional probability is weighed by the importance of a word. The weights are determined using a commonsense KB in order to provide information that humans might implicitly be reasoning about when answering such questions (Section 3.2). Following Niu et al. (2021), we expand the set of answer candidates by generating free-text answers, to increase the scorer's robustness to lexical variability (Section 3.3). An overview of the method is shown in Figure 2.

### 3.1 Basic Scoring Method

The basic scoring method directly uses the LM score, which is calculated by accumulating the conditional probabilities assigned by the LM for each token given the prefix. Given a question $Q = q_1...q_{n_Q}$ and an answer choice $A_i = a_{i,1}...a_{i,n_{A_i}}$, we convert $Q$ into a declarative statement $s$ (see Appendix A), and define the LM score of answer choice $A_i$ as follows:

$$P_{A_i} = P(A_i|s)$$
$$= \frac{1}{n_s + n_{A_i}} \cdot \prod_{j=1}^{n_{A_i}} P(a_{i,j}|s, a_{i,1}, \cdots, a_{i,j-1}) \quad (1)$$

where $n_s$ is the number of tokens in $s$.

Finally, we can determine the most plausible choice $\hat{A}$ among the answer choices based on their corresponding scores:

$$\hat{A} = \arg\max_i P_{A_i} \qquad (2)$$

## 3.2 Commonsense Augmented Scoring

The importance of individual words in the question and their contribution to choosing the correct answer varies greatly. Take for example the instance in Figure 1, taken from the COPA dataset (Gordon et al., 2012). Determining the cause of the event "The woman hired a lawyer" involves reasoning about the circumstances in which one might hire a lawyer, such as if they are suing someone. In this case, the keywords "lawyer" from the context and "sue" from the answer choice, and the semantic relation between them (i.e., suing someone requires a lawyer), supports the correct prediction. To that end, CASE first identifies important keywords from the question and answer choices (Section 3.2.1). Each keyword is assigned an importance score, and the conditional probability $P_A$ is updated by considering the importance of each token in the answer choice (Sec 3.2.2).

### 3.2.1 Keywords Extraction

Given a question $Q$ and an answer choice $A$, we use YAKE (Campos et al., 2018), an unsupervised automatic keyword extraction method, to extract a set of keywords $\text{Key}_Q \subset Q$ and $\text{Key}_A \subset A$. In particular, we are interested in finding the keywords from each answer choice that are important in the context of the question $Q$, which we denote $\text{Key}_{A|Q} \subset \text{Key}_A$. To that end, we use ConceptNet (Speer et al., 2017), a commonsense knowledge base, to find paths between terms in $\text{Key}_Q$ and $\text{Key}_A$, and include in $\text{Key}_{A|Q}$ keywords from the answer choice that are connected in ConceptNet to keywords from the question:

$$\text{Key}_{A|Q} = \left\{ a \in \text{Key}_A \,\middle|\, \begin{array}{l} \exists q \in \text{Key}_Q \,\wedge \\ \exists p = a \rightsquigarrow q \in \text{CN} \,\wedge \\ |p| \leq k \end{array} \right\} \qquad (3)$$

where $p$ denotes a path in ConceptNet (CN) with up to $k$ edges.

### 3.2.2 Weight Assigning

We assign a weight to each token $a \in \text{Key}_{A|Q}$ based on the strength of its connection to keywords in $\text{Key}_Q$. To that end, we look at all the ConceptNet paths that connect $a$ with keywords in $\text{Key}_Q$, which

we denote $\text{Paths}_{a\rightsquigarrow}$. We convert the path to a set of sentences by expressing each edge as a natural language sentence, based on relation templates (see Appendix B). For example, the path sue $\xrightarrow{\text{related to}}$ law $\xleftarrow{\text{in context of}}$ lawyer is expressed as $S_1$ = "sue is related to law" and $S_2$ = "lawyer is a word used in the context of law". We use the LM to score a single path $P_{a\rightsquigarrow q}$ as follows. First, the score $S(E_i)$ of edge $E_i = (x_i, R_i, y_i)$ is calculated as the conditional probability of generating the second node $y_i$ following the textual template of relation $R_i$, to which we assign the first node $x_i$, such as P(law|sue is related to). We use the chain rule for conditional probability to compute the score of the entire path:

$$S(P_{a\rightsquigarrow q}) = \frac{1}{|P_{a\rightsquigarrow q}|+1} \left( \sum_1^{|P_{a\rightsquigarrow q}|} \log S(E_i) + \log S(E') \right) \qquad (4)$$

where $E'$ is an artificial summary edge from $x_1$ to $y_{P_{a\rightsquigarrow q}}$ with the "is related to" relation, such as "sue is related to lawyer".

To get an aggregated score for a token $a$, we sum the scores of all paths in $\text{Paths}_{a\rightsquigarrow}$:

$$S_{\text{Paths}_{a\rightsquigarrow}} = \sum_{P_{a\rightsquigarrow q} \in \text{Paths}_{a\rightsquigarrow}} S(P_{a\rightsquigarrow q}) \qquad (5)$$

Finally, the weight for each token $a_{i,j}$ in $A_i$ is computed as follows.

$$W_{a_{i,j}} = \begin{cases} 1 + \lambda S_{\text{Paths}_{a_{i,j}\rightsquigarrow}}, & \text{if } a_{i,j} \in \text{Key}_{A_i|Q} \\ 1, & \text{if } a_{i,j} \notin \text{Key}_{A_i|Q} \end{cases} \qquad (6)$$

where $\lambda$ is a hyperparameter (§4.3).

With the weights for each token, we can now update the LM score defined in Equation 1 to a weight-based plausibility score as follows:

$$P_{A_i} = \prod_{j=1}^{n} W_{a_{i,j}} \cdot P(a_{i,j}|s, a_{i,1}, \cdots, a_{i,j-1}) \qquad (7)$$

## 3.3 Expanded Answer Space

The final addition to our model aims at reducing the LM sensitivity to the phrasing of the correct answer. For example, an infrequent word in the correct answer choice can reduce the overall probability of the choice and make the LM predict another option as more plausible (Holtzman et al., 2021). To mitigate this issue, we follow Niu et al. (2021) and expand the set of answer candidates by using a causal LM to generate open ended answers $A^* =$

$\{A_1^*, ..., A_{n_{A^*}}^*\}$. The idea is to allow the model to consider various phrasings of the same conceptual answer. For example, in Figure 2, the generated answer $C_1$ is a paraphrase of answer choice $A$.

We treat the generated answer choices $A^*$ the same as the original answer choices $A$ and compute the score for each answer $A_i^* \in A^*$ using Equation 7. To map the answer choices back into the original answer space $A$, we attempt to match each $A_i^* \in A^*$ to $A_i \in A$ based on two criteria: sentence similarity and keyword connections.

**Sentence Similarity.** We use the Sentence-Transformer package (Reimers and Gurevych, 2019) to represent the answers, and compute the cosine similarity between the representations of each generated answer in $A^*$ and original answer in $A$. The similarity score between the sentence pair should be above $s_{sim}$.

**Keyword Connections.** We calculate the connection score between the keywords in each generated answer in $A^*$ and each original answer in $A$ using the method introduced in Sec 3.2.2. We require the connection score to be greater than 0.

A candidate can only be assigned to a group if it meets both thresholds, and we discard generated answers that are not mapped into answer choices in $A$. Once we mapped generated answers to original answers, the final prediction of the model modifies Equation 2 to select the highest scores of all answers within the same cluster:

$$\hat{A} = \arg\max_i \arg\max_j P_{A_{i,j}} \qquad (8)$$

where $A_{i,j}$ is the $j$th answer in cluster $A_i$.

# 4 Experimental Setup

## 4.1 Datasets

We evaluated our method on five multiple-choice commonsense question answering datasets described below.

**COPA.** The goal in the **C**hoice **o**f **P**lausible **A**lternatives dataset (COPA; Roemmele et al., 2011) is, given a premise event, to choose the more plausible cause or effect among two alternatives.

**SCT.** The **S**tory **C**loze **T**est dataset (SCT; Mostafazadeh et al., 2016) is a collection of four-sentence stories with two possible endings. The goal is to predict which ending is more plausible following the beginning of the story.

**SocialIQA.** The **S**ocial **I**nteraction **Q**uestion **A**nswering (SocialIQA; Sap et al., 2019b) dataset tests models on their understanding of social situations and human behavior. Each question presents a hypothetical scenario followed by a question and 3 answer choices.

**ARC.** The **A**I2 **R**easoning **C**hallenge (ARC; Clark et al., 2018) consists of 7,787 science exam questions drawn from a variety of sources. The questions are divided into Easy (ARC-E) and Challenging (ARC-C) sets.

**OBQA.** The **O**pen**B**ook**QA** (OBQA; Mihaylov et al., 2018) dataset contains questions that require multi-step reasoning, use of commonsense knowledge, and rich text comprehension. The dataset has roughly 6,000 questions.

Since the test set of SCT and SocialIQA are not publicly-available, we report the accuracy on the development set for all datasets.

## 4.2 Baselines

We compare our proposed method with the basic LM-based scoring method described in Section 3.1, as well as more advanced LM-based scoring methods described below.

**Self-talk** (Shwartz et al., 2020) consists of two causal LMs. The knowledge generator LM generates clarification questions conditioned on the context and pre-defined prefixes, and their corresponding answers. The scoring LM computes the probability of each answer choice conditioned on the context and question as well as the additionally generated knowledge.[2]

**DC-PMI** (Holtzman et al., 2021) aims to eliminate the effect of the number of synonyms and the word frequency on the LM score by dividing the conditional probability (Eq 1) by a domain-conditional prior probability for the answer choice.

**SEQA** (Niu et al., 2021) uses a LM to generate a set of answer candidates. These candidates then "vote" for an original answer candidate based on their semantic similarity to each candidate, and the top-voted answer is selected as the final answer. For a fair comparison with the other model, we changed the voting model from SRoBERTa$^{NLI}$ to the origin SRoBERTa that was not further fine-tuned on an NLI dataset.

---

[2]We don't compare with follow-up work by Liu et al. (2022) since they targeted a different set of tasks.

| Methods | LM | | COPA | SCT | SocialIQA | ARC-E | ARC-C | OBQA |
|---------|-----|-----|------|-----|-----------|-------|-------|------|
| | Scoring | Generating | | | | | | |
| $LM_{sum}$ | GPT2 | - | 69.0 | 67.6 | 43.1 | 53.5 | 25.4 | 22.4 |
| $LM_{avg}$ | GPT2 | - | 68.4 | 71.5 | 45.8 | 47.4 | 28.7 | 30.8 |
| Self-talk | GPT2 | GPT2 | 66.2 | 70.4 | 47.5 | - | - | - |
| DCPMI | GPT2 | - | 70.8 | 68.6 | 39.2 | 36.0 | 25.1 | 31.4 |
| SEQA | SRoBERTa | GPT2 | 55.8 | 57.4 | 36.4 | 32.1 | 23.7 | 21.2 |
| $SEQA_{GPT3}$ | SRoBERTa | GPT3 | 66.2 | 64.4 | 40.3 | 54.4 | 34.8 | 22.2 |
| CDG | GPT2 | COMET | 72.2 | 71.5 | 45.4 | - | - | - |
| ArT | GPT2 | GPT2 | 69.8 | 71.6 | 47.3 | - | - | - |
| CAS | GPT2 | - | 70.4 | 73.0 | 46.0 | 55.8 | 28.8 | 32.6 |
| $CASE_{GPT2}$ | GPT2 | GPT2 | 73.8 | 76.1 | 46.1 | 54.4 | 30.8 | 30.2 |
| $CASE_{GPT3}$ | GPT2 | GPT3 | **78.2** | **83.2** | **48.5** | **63.2** | **36.5** | **35.2** |

Table 1: Accuracy (%) of the scoring various methods on the dev sets. All scoring methods are based on GPT-$2_{xlarge}$. $CASE_{GPT2}$ and $CASE_{GPT3}$ denote CASE with candidate generation by GPT-$2_{xlarge}$ and GPT-3 respectively. **Takeaway**: Weighting leads to substantial improvements. When combined with candidate generation, it outperforms all baselines by a large margin.

**CDG** (Bosselut et al., 2021) uses knowledge from COMET (Bosselut et al., 2019) to construct a local commonsense knowledge graph for reasoning and inference.

**ArT** (Wang and Zhao, 2022) consists of two steps: notes taking and reverse thinking. In the notes taking step, the LM generates templated inferences pertaining to key phrases in the context, which are later added as additional knowledge. The reverse thinking step aggregates the scores of different orders of the answer and question (e.g. "x because y" vs. "y therefore x").

### 4.3 Setup and Hyper-parameters

We used GPT-2 via the HuggingFace Transformers library (Wolf et al., 2020) for the scoring part, and GPT-2 XL and GPT-3 `davinci-003` for the answer space expansion step. In the keyword extraction step (§3.2.1), we included ConceptNet paths with up to $k = 3$ edges. In the weight assigning step (§3.2.2) we set the coefficient $\lambda$ to 10.

In the answer space expansion step (§3.3), we generated $n_{A^*} = 100$ answers from GPT-2 and $n_{A^*} = 50$ answers from GPT-3 for each question. Similarly to SEQA, we used nucleus sampling (Holtzman et al., 2021) with $p = 0.9$ and set a maximum length of 15 tokens for both LMs. We set the sentence similarity threshold to $s_{sim} = 0.5$ for GPT2 x-large and $s_{sim} = 0.6$ for GPT-3.

Hyper-parameter values were selected based on preliminary experiments on the training sets and were not tuned on the dev sets.

## 5 Results

### 5.1 Main Results

The performance of the various scoring methods on the 5 benchmarks are presented in Table 1. For fair comparison with the baselines, the table shows the performance when GPT2$_{xlarge}$ is used. We report the accuracy on the dev set. CAS stands for Commonsense-Augmented Scoring, i.e. it excludes the candidate generation.

The performance of CAS shows that weighting leads to substantial improvements upon the simpler baselines. CAS also stands out in the competition with DCPMI, which can also be regarded as a special weight-scoring method.

When combined with candidate generation, CASE outperforms nearly all baselines, except for the SocialIQA dataset, on which ArT and Self-talk perform better. Notably, both baselines rely on human-designed prompts to generate additional information, which might give them an advantage.

The gap in performance from SEQA, which also expands the answer space by generating candidate answers, further demonstrates the effectiveness of dynamic weighting.

### 5.2 Effect of the Scoring LM Size

Table 2 shows the performance of CAS, CASE and the simple baselines when using different sizes of GPT-2 models in the scoring part.

**Bigger is better.** Across the various methods, bigger LMs perform better than smaller LMs.

| Dataset | Methods | $\text{GPT2}_S$ | $\text{GPT2}_M$ | $\text{GPT2}_L$ | $\text{GPT2}_{XL}$ |
|---------|---------|------|------|------|------|
| COPA | $\text{LM}_{sum}$ | 60.0 | 66.6 | 69.2 | 69.0 |
| | $\text{LM}_{avg}$ | 62.6 | 65.4 | 67.0 | 68.4 |
| | CAS | 62.0 | 67.2 | 69.4 | 70.4 |
| | $\text{CASE}_{GPT2}$ | 69.6 | 72.0 | 72.2 | 73.8 |
| | $\text{CASE}_{GPT3}$ | **75.4** | **76.4** | **77.4** | **78.2** |
| SCT | $\text{LM}_{sum}$ | 58.2 | 62.7 | 64.4 | 67.9 |
| | $\text{LM}_{avg}$ | 60.4 | 66.4 | 68.8 | 71.5 |
| | CAS | 61.9 | 67.5 | 70.9 | 73.0 |
| | $\text{CASE}_{GPT2}$ | 74.0 | 75.2 | 75.7 | 76.1 |
| | $\text{CASE}_{GPT3}$ | **76.7** | **78.6** | **79.0** | **83.2** |
| SIQA | $\text{LM}_{sum}$ | 39.7 | 41.4 | 42.0 | 43.1 |
| | $\text{LM}_{avg}$ | 41.8 | 44.1 | 44.9 | 45.8 |
| | CAS | 42.8 | 44.6 | 45.7 | 46.0 |
| | $\text{CASE}_{GPT2}$ | 43.9 | 43.7 | 44.1 | 44.5 |
| | $\text{CASE}_{GPT3}$ | **47.6** | **48.4** | **48.5** | **48.5** |
| ARC-E | $\text{LM}_{sum}$ | 44.2 | 48.8 | 50.4 | 53.5 |
| | $\text{LM}_{avg}$ | 37.9 | 40.2 | 45.1 | 47.4 |
| | CAS | 46.1 | 49.8 | 53.0 | 55.8 |
| | $\text{CASE}_{GPT2}$ | 46.5 | 49.6 | 52.0 | 54.4 |
| | $\text{CASE}_{GPT3}$ | **54.2** | **59.1** | **60.0** | **63.2** |
| ARC-C | $\text{LM}_{sum}$ | 19.7 | 23.1 | 22.7 | 25.4 |
| | $\text{LM}_{avg}$ | 23.4 | 23.7 | 25.4 | 28.7 |
| | CAS | 26.4 | 26.4 | 27.4 | 28.8 |
| | $\text{CASE}_{GPT2}$ | 28.1 | 29.4 | 27.8 | 30.8 |
| | $\text{CASE}_{GPT3}$ | **33.4** | **35.3** | **33.8** | **36.5** |
| OBQA | $\text{LM}_{sum}$ | 16.2 | 18.2 | 21.8 | 22.4 |
| | $\text{LM}_{avg}$ | 23.0 | 26.8 | 30.0 | 30.8 |
| | CAS | 25.6 | 28.6 | 31.4 | 32.6 |
| | $\text{CASE}_{GPT2}$ | 26.0 | 26.6 | 27.4 | 30.2 |
| | $\text{CASE}_{GPT3}$ | **32.2** | **35.4** | **37.4** | **35.2** |

Table 2: Accuracy when using GPT2 models with different sizes for the scoring. **Takeaways**: CAS consistently outperforms standard LM scoring methods, and is outperformed by CASE. For CASE, the best performance is achieved when using large GPT2 models for scoring and more importantly, GPT3 for candidate generation.

**Smaller LMs gain more from candidate generation.** While all LMs benefit from weighting and candidate generation, smaller LMs gain bigger improvements. For example, candidate generation with GPT-3 adds 13.4 points on COPA to a $\text{GPT2}_S$ CAS scorer, but only 8.2 points for $\text{GPT2}_{XL}$. We hypothesize that the model performance is more sensitive to the LM quality when a single sentence is considered, while expanding the answer space makes even the lower-quality LMs more robust.

### 5.3 Effect of the No. of Generated Candidates

Figure 3 shows the effect of the number of generated candidates on the performance, focusing on COPA. We summarize the findings below.

**Generating more candidates leads to higher accuracy.** When generating few (< 20) candidates, the model's performance is unstable and relatively low. This might happen due to the generated answers being conceptually different from the original candidate answers, in which case they might not meet the mapping thresholds in Section 3.3 and

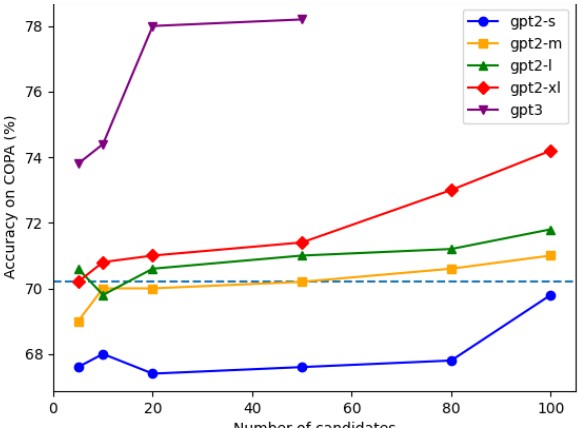

Figure 3: Accuracy curve of CASE on the COPA dev set, with different numbers of candidates generated from various LMs. The dotted line represents the baseline method $\text{LM}_{sum}$ which uses $\text{GPT2}_{xlarge}$. **Takeaways**: Generating more candidates leads to higher accuracy, but larger scoring LMs require fewer candidates.

be filtered out. This means that CASE effectively degenerates to CAS. Thus, it's important to generate a large number of candidates. This reassesses the findings in Niu et al. (2021).

**Larger models require fewer candidates.** Larger LMs generate higher quality text which is more likely to be fluent, relevant to the context, logically correct, and consistent with commonsense knowledge. Therefore, we can expect fewer candidates to be filtered out. In addition, the generated candidates may be conceptually similar and better phrased than the original choice.

### 5.4 Effect of the Weighting Strategy

Table 3 compares the COPA performance of different weighting strategies. Two baselines, $\text{LM}_{sum}$ and $\text{LM}_{avg}$, already introduced in Section 3.1, treat all tokens equally, summing or averaging the token-level probabilities. Conversely, the static weighting strategy (SW and SWC, with or without candidate generation), assigns a static number (1.5) to each selected key token. Finally, the dynamic weighting strategies (CAS and CASE) not only distinguish key tokens from unimportant ones but also assign different scores to each key token based on its semantic relevance to the question.

The results show that while the static weighting strategy outperforms the baseline when no additional candidates are generated (SW vs. LM), these strategies perform similarly when additional candidates are generated (SWC vs. LM+c). In both cases,

| | GPT2$_s$ | GPT2$_m$ | GPT2$_l$ | GPT2$_{xl}$ |
|---|---|---|---|---|
| LM$_{sum}$ | 60.0 | 66.6 | 69.2 | 69.0 |
| + SW | 61.2 | 66.6 | 70.0 | 69.6 |
| + CAS | 62.0 | 67.2 | 69.4 | 70.4 |
| + C | 69.2 | 71.8 | 70.4 | 72.4 |
| + SWC | 69.2 | 71.2 | **72.4** | 72.2 |
| + CASE | **69.6** | **72.0** | 72.2 | **73.8** |

Table 3: Accuracy on the COPA dev set when using different weight-assigning methods. The methods below the dotted line expand the answer space by generating additional answer candidates. **Takeaway**: keyword selection improves the performance, especially when it is informed by commonsense knowledge.

the static weighting strategy underperforms compared to the dynamic strategy. This result confirms that commonsense knowledge can help inform the model about the keywords that are important *for the current question*.

## 6 Qualitative Analysis

We focus on CASE and look at the individual token scores and corresponding ConceptNet paths to better understand the model decision-making process.

Figure 4 shows an example from SCT where CASE predicted the correct answer. The word "upset" in the correct answer choice was assigned a high weight by CASE thanks to ConceptNet paths such as upset $\xleftrightarrow{\text{related to}}$ depression $\xleftarrow{\text{causes}}$ stress $\xleftrightarrow{\text{related to}}$ work.

Conversely, in Figure 5, CASE predicted the incorrect answer choice for another SCT example. The model focused on the word "left" due to its semantic relation to the word "drove", failing to understand that Priya drove *to* and not *away from* the restaurant.

## 7 Conclusion

We presented CASE, a novel LM-based plausibility score for zero-shot MCQA tasks. CASE uses a commonsense KB to assign importance weights to words in the input. The weighting strategy outperforms basic LM scoring methods. When combined with generating additional answer candidates, CASE outperforms the baselines on 5 popular MCQA benchmarks. We further showed that the two approaches are complementary and are especially beneficial when using smaller LMs. In the future, we plan to explore a more selective approach

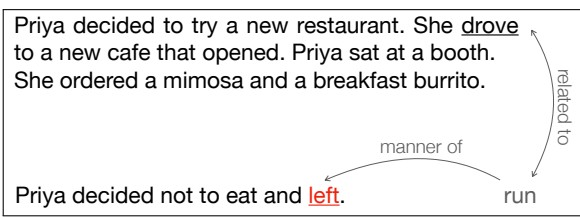

Figure 4: An SCT example, along with the correct answer predicted by CASE, and an example ConceptNet path that increased the weight of the important word *upset*.

Figure 5: A incorrectly-predicted SCT example, along with the incorrect answer predicted by CASE, and an example ConceptNet path that increased the weight of the word *left*.

for knowledge retrieval from the KB, and adapt CASE for additional NLP tasks.

## Limitations

**Computational complexity.** CASE is more computationally expensive than using a basic LM score, as it involves finding relevant paths from an external knowledge base and then estimating their likelihood with a LM, in order to gauge the importance of keywords.

**Concept coverage.** The weight assignment strategy in CASE is based on ConceptNet. The knowledge in KBs such as ConceptNet is not contextualized, which means that some facts pertaining to concepts in the instance might not be relevant to the specific context. In addition, it has limited coverage. COMET (Hwang et al., 2021) has been used in prior work (Majumder et al., 2020; Chakrabarty et al., 2020; Ravi et al., 2023) to overcome this limitation. However, finding relevant paths using COMET requires an iterative multi-hop reasoning approach (Arabshahi et al., 2021) which is more complex, and more computationally-intensive. We aim to explore efficient ways to achieve this in future work.

**Answer format.** Since our method assigns a weight for each word in the input, it is only ap-

plicable for MCQA tasks in which the answer is a sentence. The weighting would be trivial for tasks with single word answers such as CommonsenseQA (Talmor et al., 2019) and BoolQ (Clark et al., 2019).

**Performance limit.** Our model demonstrates a significant performance improvement over other zero-shot baselines across a majority of datasets. However, it is worth noting that the state-of-the-art performance on the datasets in this paper is achieved with more supervision (i.e. supervised or few-shot models).

## Ethics Statement

**Data.** All the datasets and knowledge bases used in this work are publicly available. We used ConceptNet as a source of commonsense knowledge. Since ConceptNet was crowdsourced, some of the knowledge may contain societal biases or prejudices held by the annotators (Mehrabi et al., 2021).

**Models.** The GPT-2 models are publicly accessible via HuggingFace, while GPT-3 is a closed model behind an API. All language models may generate offensive statements if prompted with specific inputs, however, our model only generates text internally while the end output is a choice between human-written answer candidates.

## Acknowledgements

This work was funded, in part, by the Vector Institute for AI, Canada CIFAR AI Chairs program, an NSERC discovery grant, and a research gift from AI2.

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

# A  Question Prompts

Table 4 shows the prompts used for each dataset. For tasks with several specific question type such as COPA and SocialIQa, we convert each question type to a natural language proxy following previous work (e.g. Shwartz et al., 2020). For tasks that present an open-ended question, we append the prefix "The answer is". Finally, for tasks that are already designed to expect the next word or sentence (such as SCT), we use the instance as is.

| Dataset | Question |
|---|---|
| COPA | My body cast a shadow over the grass [because] |
| | The physician misdiagnosed the patient [so] |
| SCT | Tyler went to a baseball game. He saw his favorite team! His team played hard. His team won! [] |
| SocialIQa | Tracy didn't go home that evening and resisted Riley's attacks. [Before, Tracy needed to] |
| ARC | Which technology was developed most recently? [the answer is] |
| | A green plant absorbs light. A frog eats flies. These are both examples of how organisms [] |
| OBQA | A person can grow cabbage in January with the help of what product? [the answer is] |
| | Gas can fill any container it is given, and liquid [] |

Table 4: Question formats used for each dataset. The red words in square brackets are additions to the context, designed specifically for each dataset.

# B  Relation Templates

Table 5 displays the templates we used to convert edges with different relation types in ConceptNet to natural language sentences, following Davison et al. (2019).

| Relation Type | Template |
|---|---|
| $A \xleftarrow{\text{related to}} B$ | A is related to B |
| $A \xrightarrow{\text{form of}} B$ | A is a form of B |
| $A \xrightarrow{\text{is a}} B$ | A is a B |
| $A \xrightarrow{\text{part of}} B$ | A is a part of B |
| $A \xrightarrow{\text{has a}} B$ | A has a B |
| $A \xrightarrow{\text{used for}} B$ | A is used for B |
| $A \xrightarrow{\text{not used for}} B$ | A is not used for B |
| $A \xrightarrow{\text{capable of}} B$ | A is capable of B |
| $A \xrightarrow{\text{not capable of}} B$ | A is not capable of B |
| $A \xrightarrow{\text{at location}} B$ | A is a location for B |
| $A \xrightarrow{\text{causes}} B$ | A causes B |
| $A \xrightarrow{\text{has subevent}} B$ | B happens as a subevent of A |
| $A \xrightarrow{\text{has first subevent}} B$ | A begins with B |
| $A \xrightarrow{\text{has last subevent}} B$ | A ends with B |
| $A \xleftarrow{\text{has prerequisite}} B$ | B is a dependency of A |
| $A \xrightarrow{\text{has property}} B$ | A can be described as B |
| $A \xrightarrow{\text{not has property}} B$ | A can not be described as B |
| $A \xrightarrow{\text{motivated by goal}} B$ | Someone does A because they want result B |
| $A \xrightarrow{\text{obstructed by}} B$ | A is a obstacle in the way of B |
| $A \xrightarrow{\text{desires}} B$ | A desires B |
| $A \xrightarrow{\text{not desires}} B$ | A do not desire B |
| $A \xrightarrow{\text{created by}} B$ | A is created by B |
| $A \xrightarrow{\text{synonym}} B$ | A is similar to B |
| $A \xleftarrow{\text{antonym}} B$ | A is opposite to B |
| $A \xrightarrow{\text{distinct from}} B$ | A is distinct from B |
| $A \xrightarrow{\text{derived from}} B$ | A is derived from B |
| $A \xrightarrow{\text{symbol of}} B$ | A is a symbol of B |
| $A \xrightarrow{\text{defined as}} B$ | A is defined as B |
| $A \xrightarrow{\text{manner of}} B$ | A is a specific way to do B |
| $A \xleftarrow{\text{located near}} B$ | A is near to B |
| $A \xrightarrow{\text{has context}} B$ | A is a word used in the context of B |
| $A \xleftarrow{\text{similar to}} B$ | A is similar to B |
| $A \xleftarrow{\text{etymologically related to}} B$ | A have a common origin with B |
| $A \xrightarrow{\text{etymologically derived from}} B$ | A is derived from B |
| $A \xrightarrow{\text{causes desire}} B$ | A makes someone want B |
| $A \xrightarrow{\text{made of}} B$ | A is made of B |
| $A \xleftarrow{\text{receives action}} B$ | B can be done to A |

Table 5: Natural language templates for each relation type in ConceptNet.