# OpenReview forum: "CASE: Commonsense-Augmented Score with an Expanded Answer Space"
_EMNLP/2023/Conference — EMNLP 2023 Findings_

### Official Review · Reviewer_8bm5 · 2023-08-03

**Soundness:** 4

**Excitement:**

2: Mediocre: This paper makes marginal contributions (vs non-contemporaneous work), so I would rather not see it in the conference.

**Missing References:**

Relevant zero-shot CSQA methods:
[1] Knowledge-driven Data Construction for Zero-shot Evaluation in Commonsense Question Answering, 2020
[2] Zero-shot Commonsense Question Answering with Cloze Translation and Consistency Optimization, 2021
[3] CAR: Conceptualization-Augmented Reasoner for Zero-Shot Commonsense Question Answering, 2023 (optional, as it's contemporaneous)

**Paper Topic And Main Contributions:**

The main contributions of this paper are the introduction of CASE, a Commonsense-Augmented Score with an Expanded Answer Space, which improves the performance of language models on multiple-choice QA tasks. CASE reduces noise from unimportant words and informs the model of implicit commonsense knowledge that may be useful for answering the question. Additionally, by generating lexically divergent answers that are conceptually similar to the choices, CASE expands the answer space and outperforms strong baselines on 5 commonsense benchmarks. The main contribution type of this paper lies in the scope of NLP Engineering experiments.

**Reasons To Accept:**

The proposed framework appears to be well thought-out, with an interesting motivation behind it. The authors have effectively integrated the score of each answer and utilized vanilla pretrained language models, including large language models such as GPT3. The experiment results demonstrate the superiority of CASE in comparison to a diverse range of zero-shot QA systems. The comparison is comprehensive, covering most of the prior methods that have been discussed. The analyses are well-designed and reflective, providing valuable insights into the performance of the proposed framework.

**Reasons To Reject:**

- First, while the proposal of a new token-level weighting method to consider the importance of different words within a sentence is effective, other zero-shot QA pipelines, such as fine-tuning models on knowledge bases or graphs, have stronger performance than the author's proposed method. Additionally, with the recent emergence of large language models that are strong QA solvers, the impact of the proposed method towards the community may be limited. While I do not reject the proposed method, I believe it is more likely to be incremental in the vanilla LM scoring pipeline compared to other zero-shot QA methods or models. It might be interesting to explore using the generated embeddings of GPT-3 to strengthen the scoring step to enhance the added value of this paper in the current NLP era.
- There are some confusing notations in Table 1 and 2, where a frequently appeared baseline, LM, is not explicitly stated. I assume it is GPT-2 (from line 417), but there is no clear instructions.
- I appreciate the idea of answer set expansion, and it makes sense that it is effective. However, I suggest that the authors conduct a further analysis regarding the quality of the generated answers. For example, they could assess how many of the generations are plausible and how many are correct by using expert annotation on a sample of the generated datasets. Furthermore, it would be beneficial to show that sentence similarity and keyword connections are playing their expected roles when linking the generations.


**Reproducibility:**

3: Could reproduce the results with some difficulty. The settings of parameters are underspecified or subjectively determined; the training/evaluation data are not widely available.

**Reviewer Confidence:**

4: Quite sure. I tried to check the important points carefully. It's unlikely, though conceivable, that I missed something that should affect my ratings.

**Typos Grammar Style And Presentation Improvements:**

In the final sentence of the abstract, the author states, "when using smaller LLMs." I believe it would be more appropriate to say, "when using smaller LMs." Additionally, in the experimental section, the author employs different terms to refer to the same approach, using "answer expansion" and "candidate generation" interchangeably. This inconsistency may lead to confusion during the initial reading.

---

> ### Author Rebuttal · Authors · 2023-08-28
>
> We thank the reviewer for the helpful feedback.
>
> **First, while the proposal of a new token-level weighting method to consider the importance of different words within a sentence is effective, other zero-shot QA pipelines, such as fine-tuning models on knowledge bases or graphs, have stronger performance than the author's proposed method.**
>
> This is a good question. While fine-tuning the LM on a knowledge base may lead to higher performance, it would be at a significant computational cost compared to our zero-shot model. Additionally, by accessing the KB during inference, our method doesn’t need to be re-trained to adapt to changes in the KB. Furthermore, our method provides interpretability. By looking at  the token-level weights (Figure 1) and the ConceptNet paths (Figures 4 and 5), we are able to interpret the model’s decision making and the knowledge that was used during the process. This would be less straightforward for LMs fine-tuned on a KB..
>
> **Additionally, with the recent emergence of large language models that are strong QA solvers, the impact of the proposed method towards the community may be limited. While I do not reject the proposed method, I believe it is more likely to be incremental in the vanilla LM scoring pipeline compared to other zero-shot QA methods or models. It might be interesting to explore using the generated embeddings of GPT-3 to strengthen the scoring step to enhance the added value of this paper in the current NLP era.**
>
> Those are great ideas for future work!
>
> **There are some confusing notations in Table 1 and 2, where a frequently appeared baseline, LM, is not explicitly stated. I assume it is GPT-2 (from line 417), but there is no clear instructions.**
>
> GPT2 in Tables 1 and 2 is GPT-2-xlarge, as we mentioned in line 417. We will mention it explicitly in the captions of the two tables.
>
> **I appreciate the idea of answer set expansion, and it makes sense that it is effective. However, I suggest that the authors conduct a further analysis regarding the quality of the generated answers. For example, they could assess how many of the generations are plausible and how many are correct by using expert annotation on a sample of the generated datasets. Furthermore, it would be beneficial to show that sentence similarity and keyword connections are playing their expected roles when linking the generations.**
>
> Thank you for recognizing our idea! We will try to include at least a small-scale annotation of the quality of generated answers in the camera-ready version.
>
> We will fix all the typos in the camera-ready version.

---

### Official Review · Reviewer_pNxV · 2023-08-04

**Typos Grammar Style And Presentation Improvements:** n/a
**Soundness:** 4

**Excitement:**

4: Strong: This paper deepens the understanding of some phenomenon or lowers the barriers to an existing research direction.

**Missing References:**

n/a

**Paper Topic And Main Contributions:**

The authors present a new technique for LM-based scoring, evaluated on multiple choice QA (MCQA). Typically, LM-based scoring is done by measuring the joint probability of LM-based next token probabilities (i.e. summing the log probabilities) and optionally applying length normalization. As others have demonstrated, this approach has weaknesses in the zero-shot setting especially due to bias towards specific tokens (as discussed in Holtzman et al). To address this weakness, the authors introduce a re-weighting scheme that assigns higher or lower weight to tokens in the answer based on their relevance to the question. To assign relevance weights, the authors use ConceptNet and measure the path length between concepts in the question and answer.

**Questions For The Authors:**

There is mention that few-shot and supervised sota is higher for these tasks. How much higher? Perhaps it is worth including these numbers in the appendix. Ideally, including results for GPT2 and/or GPT3.

**Reasons To Accept:**

A1. The re-weighting approach using conceptnet is clever and leads to nice results. It's also encouraging to see this is complementary to answer expansion.

A2. The main experiments are on a large number of datasets (5 datasets) and includes multiple baselines. There is extensive analysis on model size and ablations for weight assignment.

A3. The paper is upfront about some limitations.

**Reasons To Reject:**

R1. The authors claim to make fair comparison to previous baselines, although I would consider using ConceptNet to be an advantage. This is most blatant when the authors the SEQA baseline that leverages NLI. Perhaps it should be explained more why using NLI is considered unfair but ConceptNet is fair game.

R2. As the authors mentioned, conceptnet is somewhat limited in coverage especially compared with the training data of modern LMs. Perhaps there is a simple extension of this work that utilizes word embeddings and/or model probabilities and doesn't need to rely on a KB like Conceptnet. Even using a bag-of-words-based reweighting would be an informative baseline and/or alternative. That being said, seeing good results from conceptnet alone is encouraging.

R3. The approach works for condition probabilities of the answer, but it's not clear if it would work for conditional probability of the question.

**Reproducibility:**

4: Could mostly reproduce the results, but there may be some variation because of sample variance or minor variations in their interpretation of the protocol or method.

**Reviewer Confidence:**

4: Quite sure. I tried to check the important points carefully. It's unlikely, though conceivable, that I missed something that should affect my ratings.

---

> ### Author Rebuttal · Authors · 2023-08-28
>
> We thank the reviewer for the helpful feedback.
>
> **The authors claim to make fair comparison to previous baselines, although I would consider using ConceptNet to be an advantage. This is most blatant when the authors the SEQA baseline that leverages NLI. Perhaps it should be explained more why using NLI is considered unfair but ConceptNet is fair game.**
>
> Thanks for bringing this up, this is a fair point. Our distinction was for zero-shot vs. supervised methods, and we considered pre-training the LM on NLI datasets an unfair advantage compared to using an off-the-shelf LM. With that said, we will include the original SEQA results, which are comparable with CASE, in the paper. We will also include a version of CASE that is fine-tuned on an NLI dataset for complete comparison.
>
> **As the authors mentioned, conceptnet is somewhat limited in coverage especially compared with the training data of modern LMs. Perhaps there is a simple extension of this work that utilizes word embeddings and/or model probabilities and doesn't need to rely on a KB like Conceptnet. Even using a bag-of-words-based reweighting would be an informative baseline and/or alternative. That being said, seeing good results from conceptnet alone is encouraging.**
>
> Thanks for the suggestions! Definitely, we agree there is room for improvement and acknowledge the limitations of using ConceptNet directly (i.e., lack of coverage and contextualization). We plan to work on more advanced methods in the future.
>
> **The approach works for condition probabilities of the answer, but it's not clear if it would work for conditional probability of the question.**
>
> That’s a good question. Although it’s not immediately clear to us what would be the advantage of measuring the probability of the question instead of the answer, we will investigate this.
>
> **There is mention that few-shot and supervised sota is higher for these tasks. How much higher? Perhaps it is worth including these numbers in the appendix. Ideally, including results for GPT2 and/or GPT3.**
>
> That’s a good idea, we will  add them to the camera-ready version.

---

### Official Review · Reviewer_sJkV · 2023-08-10

**Soundness:** 3

**Excitement:**

4: Strong: This paper deepens the understanding of some phenomenon or lowers the barriers to an existing research direction.

**Paper Topic And Main Contributions:**

This article primarily introduces a rationality score for a zero-shot Multiple Choice Question Answering (MCQA) task based on a commonsense knowledge base, known as CASE. This score addresses the limitation of treating all words equally important by assigning importance weights to words in the input. CASE reduces the noise from less important words through dynamic weight assignment, and leverages implicit commonsense knowledge for answering questions. Furthermore, the article proposes a method to expand the answer space by generating diverse yet conceptually similar answers at the lexical level, aiming to enhance the model's performance. Across five common commonsense benchmark tests, the proposed approach outperforms baseline models. Additionally, the article demonstrates the complementarity of these two methods, especially when using a smaller language model.

**Questions For The Authors:**

I have examined the results report in the SEQA paper (https://aclanthology.org/2021.acl-long.237.pdf), but upon comparison, I have observed discrepancies between the reported results in the current paper and those presented in the SEQA paper's Table 2.

**Reasons To Accept:**

1. This paper proposed an effective method called CAS, which has demonstrated its efficacy in the context of GPT-based answer selection tasks.
2. Extensive experiments has been undertaken to validate the effectiveness of CAS, besides, it yields noteworthy enhancements in the performance of the previously established SEQA method.
3. The presented methodology holds a broad applicability, capable of integration into diverse architectures, the intuition behind the method is sound.

**Reasons To Reject:**

1. The scope of the proposed method is conventional and constrained, as it is solely applicable to answer selection tasks, without extending its utility to the enhancement of answer generation tasks.
2. Furthermore, the applicability of the proposed method within a few-shot setting remains a pertinent consideration. The inclusion of experiments conducted within a few-shot scenario would be advisable, as it has the potential to yield practical advantages for real-world applications.
3. Notably, GPT-3 davinci-003 serves as the basis for answer space expansion. However, it is noteworthy that a comprehensive evaluation of the performance of GPT-3 davinci-003 is absent from the reported findings.

**Reproducibility:**

4: Could mostly reproduce the results, but there may be some variation because of sample variance or minor variations in their interpretation of the protocol or method.

**Reviewer Confidence:**

3: Pretty sure, but there's a chance I missed something. Although I have a good feel for this area in general, I did not carefully check the paper's details, e.g., the math, experimental design, or novelty.

---

> ### Author Rebuttal · Authors · 2023-08-28
>
> We thank the reviewer for the helpful feedback.
>
> **The scope of the proposed method is conventional and constrained, as it is solely applicable to answer selection tasks, without extending its utility to the enhancement of answer generation tasks.**
>
> We note that many NLP datasets are designed as multiple-choice question answering tasks, hence our method is broadly applicable to many NLP datasets. While adapting our method to generative tasks is beyond the scope of this paper, we hypothesize that it would be straightforward to adapt it by treating the answer expansion phase as a way to generate the initial answer choices. With that said, generative tasks are harder to evaluate automatically, so we leave this for future work.
>
>
> **Furthermore, the applicability of the proposed method within a few-shot setting remains a pertinent consideration. The inclusion of experiments conducted within a few-shot scenario would be advisable, as it has the potential to yield practical advantages for real-world applications.
> Notably, GPT-3 davinci-003 serves as the basis for answer space expansion. However, it is noteworthy that a comprehensive evaluation of the performance of GPT-3 davinci-003 is absent from the reported findings.**
>
> First, our paper focuses on the zero-shot setting, however, we acknowledge the strengths of the few-shot setup and we plan to explore extending our method to this setup in the future. Second, we note that our model requires access to the individual token probabilities so it only supports open models or models that provide this information.
>
>
> **I have examined the results report in the SEQA paper (https://aclanthology.org/2021.acl-long.237.pdf), but upon comparison, I have observed discrepancies between the reported results in the current paper and those presented in the SEQA paper's Table 2.**
>
> The results in the SEQA paper are better because they used sentence-RoBERTa, which is fine-tuned on NLI datasets. For this reason, SEQA doesn't strictly conform to the zero-shot setting. Therefore, we re-ran the SEQA experiments with the original sentence-RoBERTa model. With that said, we will include the original SEQA results, which are comparable with CASE, in the paper. We will also include a version of CASE that is fine-tuned on an NLI dataset for complete comparison.

---

### Official Review · Reviewer_CJBH · 2023-08-12

**Typos Grammar Style And Presentation Improvements:** 1. In Equation 1, should $s$ should b…
**Soundness:** 3

**Excitement:**

3: Ambivalent: It has merits (e.g., it reports state-of-the-art results, the idea is nice), but there are key weaknesses (e.g., it describes incremental work), and it can significantly benefit from another round of revision. However, I won't object to accepting it if my co-reviewers champion it.

**Paper Topic And Main Contributions:**

This paper addresses problems arising from scoring sequences of text using token-level probabilities output by a language model in the context of multiple-choice question answering. In particular, the authors make the claim that the performance of standard sequence scoring is negatively affected due to equal weights being assigned to all tokens. To overcome this problem, this work proposes a dynamic importance-weighting scheme that increases the probability scores of words that are deemed to be more relevant to the task/question at hand. The signal for importance, or relevance, is based on word-level scores derived from descriptions contained in the ConceptNet KG. Lastly, the authors propose LM-based answer space expansion to generate additional answers, which are then semantically clustered and mapped back to the original answer space.

**Questions For The Authors:**

1. In section 3.2.1, it is unclear what the automatic keyword extraction is optimizing for when picking the initial keywords before the paths are searched for in ConceptNet. Could you clarify?

**Reasons To Accept:**

The paper presents a novel common-sense augmented re-weighting scheme couple with candidate answer expansion that shows positive empirical performance in the task of multi-choice question answering.

**Reasons To Reject:**

While interesting, the results seem to suggest that the majority of the gains are had due to the candidate expansion rather than the common-sense re-weighting scheme. Further, it is unclear if this method generalizes beyond GPT-2 to newer low-to-moderate-sized LLMs. Evaluations with multiple models would be interesting to see.

**Reproducibility:**

3: Could reproduce the results with some difficulty. The settings of parameters are underspecified or subjectively determined; the training/evaluation data are not widely available.

**Reviewer Confidence:**

3: Pretty sure, but there's a chance I missed something. Although I have a good feel for this area in general, I did not carefully check the paper's details, e.g., the math, experimental design, or novelty.

---

> ### Author Rebuttal · Authors · 2023-08-28
>
> We thank the reviewer for the helpful feedback.
>
> **While interesting, the results seem to suggest that the majority of the gains are due to the candidate expansion rather than the common-sense re-weighting scheme.**
>
> As we show in Table 1, when candidate expansion and commonsense-reweighting are combined, they lead to outperforming the baselines by a large margin. While the gains from candidate expansion are bigger, Table 1 shows that each of these components yields improvements, in particular the gap in performance from the LM base scores shows the benefit of weighting. These two components complement each other.
>
> **Further, it is unclear if this method generalizes beyond GPT-2 to newer low-to-moderate-sized LLMs. Evaluations with multiple models would be interesting to see.**
>
> Thanks for the suggestion to test our method on multiple models. We will add baselines based on other low/moderate sized LLMs (e.g. T5, BART) in the camera-ready version.
>
> **In section 3.2.1, it is unclear what the automatic keyword extraction is optimizing for when picking the initial keywords before the paths are searched for in ConceptNet. Could you clarify?**
>
> We used YAKE, an unsupervised keyword extractor, to extract salient words from a document before we search paths on ConceptNet. Yake doesn’t rely on any external corpus or dictionary, and it is based on features such as casing, position, word frequency, and word relatedness to context. Our motivation was to only focus on words that provide important information for the sentence, and to reduce the overall search time. We will clarify this in the paper.
>
> **In L257, S is being used to denote the scoring function while the same symbol was used in L199 to denote the declarative statement.**
>
> Thanks for pointing this out, it is indeed confusing, and we will change the notation in the camera-ready version.
>
> **In Equation 1, is there a reason for operating in the probability space instead of the more typical log probability space?**
>
> Thanks for pointing this out. We wrote the equation in the probability space for the sake of simplicity, but in practice all computations were done in the log space for numerical stability. We will clarify it in the paper.

---

### Meta-Review · Area_Chair_qbz4 · 2023-09-23

**Recommendation:** 4

**Metareview:**

The paper tackles the issue of answer scoring based on LM probabilities for common-sense related tasks. The main idea is to use a weighting sum of answer token probabilities, where the weights are derived by path-based scores calculated from ConceptNet. The answer set is also expanded to allow for better scoring.

The problem and solution are well-motivated. The main issue of obtaining answer probabilities via an aggregation of token probabilities is an important one. The proposed method of finding weights on the answer token probabilities is reasonable.  The experimental evidence is mostly sound with some minor caveats. The experiments were are all done on GPT-2 a relatively weaker model compared more recent ones (e.g. T5, UnifiedQA) of similar sizes. The paper uses ConceptNet as extra information during answer scoring. A tighter claim on answer token weighting would have been to include ConceptNet paths via other means in the model (e.g. as inputs to the model).  There are adequate analyses of the components in the paper. The writing is clear overall.

The work has significance within the scope of selection technique in common-sense QA tasks. The main contribution is in showing that weighting token probabilities helps for answer selection. This is a neat idea that is relatively simple to implement and use. Another plus for significance is that it adds to methods that tackle calibration of LM probabilities, which allow us to get more out of LMs when used as black-boxes.

I am less excited about the broader significance of the work for the following reason. The overall idea seems similar to rescoring methods in generation tasks, where a base LM generates multiple candidates and some scoring method rescores the outputs, which can also be integrated into the beam search. It is not clear if there was some new challenge or twist to address in implementing the idea for answer selection.

---

### Decision · Program_Chairs · 2023-10-07

**Decision:**

Accept-Findings

**Comment:**

The paper tackles the issue of answer scoring based on LM probabilities for common-sense related tasks. The main idea is to use a weighting sum of answer token probabilities, where the weights are derived by path-based scores calculated from ConceptNet. The answer set is also expanded to allow for better scoring.

The problem and solution are well-motivated. The main issue of obtaining answer probabilities via an aggregation of token probabilities is an important one. The proposed method of finding weights on the answer token probabilities is reasonable.  The experimental evidence is mostly sound with some minor caveats. The experiments were are all done on GPT-2 a relatively weaker model compared more recent ones (e.g. T5, UnifiedQA) of similar sizes. The paper uses ConceptNet as extra information during answer scoring. A tighter claim on answer token weighting would have been to include ConceptNet paths via other means in the model (e.g. as inputs to the model).  There are adequate analyses of the components in the paper. The writing is clear overall.

The work has significance within the scope of selection technique in common-sense QA tasks. The main contribution is in showing that weighting token probabilities helps for answer selection. This is a neat idea that is relatively simple to implement and use. Another plus for significance is that it adds to methods that tackle calibration of LM probabilities, which allow us to get more out of LMs when used as black-boxes.

I am less excited about the broader significance of the work for the following reason. The overall idea seems similar to rescoring methods in generation tasks, where a base LM generates multiple candidates and some scoring method rescores the outputs, which can also be integrated into the beam search. It is not clear if there was some new challenge or twist to address in implementing the idea for answer selection.